# Modeling Stygofauna Resilience to the Impact of the Climate Change in the Karstic Groundwaters of South Italy

Agostina Tabilio Di Camillo and Costantino Masciopinto *

Consiglio Nazionale delle Ricerche, Istituto di Ricerca Sulle Acque, Viale Francesco De Blasio, 5, 70132 Bari, Italy
* Correspondence: costantino.masciopinto@ba.irsa.cnr.it

**Abstract:** We predicted the global warming effects on the stygofauna of Murgia–Salento karstic groundwaters in Italy for 2050, which contribute to a biodiversity loss assessment in the climate change context. For quantitative impact estimations, we defined a local resilience score (LRS) for sampled species between 2018 and 2021. A resilience model equation of the stygobiont species conservation was obtained from a surface best-fit of the assigned LRS and the corresponding values of independent variables describing the environmental quality of monitored habitats and LRS. The principal components of the correlation between the monitored variables and LRS were obtained via factor analysis. Three-dimensional surface maps of stygofauna species resilience (SSR) were constructed to visualize and quantitatively compare the biodiversity loss of species assemblages owing to environmental and habitat quality modifications. The proposed SSR model was applied to the sampled stygofauna, and the decrease in local species resilience for 2050 was predicted. Independent variable factors were updated for 2050 to consider increases of up to 2 °C and 0.04 mS/cm in groundwater temperature and electric conductance observed for 2021. The SSR model results predicted a high impact on the resilience of *Parastenocaris* cf. *orcina* (80%), newly retrieved Crustacea Copepod Cyclopidae gen 1 sp 1, and three other stygobites (~50%). The resilience of *Metacyclops stammeri* had minor impacts.

**Keywords:** adaptive resilience models; biodiversity loss; climate change; groundwater ecosystem; karstic stygofauna; species resilience; taxa replacement

## 1. Introduction

Groundwater comprises a unique ecosystem largely composed of groundwater-adapted marine crustacean species, such as mysids, amphipods, and copepods, which provide useful ecosystem services by recycling wastewater infiltration into groundwater. However, anthropogenic activities and global warming impose increased stress on underground biota worldwide [1].

Karstified aquifers and related hypogeous habitats host a higher troglofaunal and stygofaunal biodiversity compared to the non-karstified groundwater owing to the availability of habitats in niches and cavities [2]. Stygofauna include animals that exclusively live in groundwater and are mainly small crustaceans ranging from 0.3 to 10 mm in length, other than fishes, worms, snails, mites, and insects. Troglofauna are air-breathing terrestrial taxa that live exclusively in dark underground caves and small air-filled niches [3]. Many karst regions are classified as hotspots of biodiversity because of the variety of species observed in different habitats in surface water (ponds and lakes), epikarst water (cavities, fractures, and conduits), unsaturated fissures (vadose zones), and deep groundwater fractures and conduits [4]. Zagmajster et al. [5], according to the Linnean biodiversity shortfall [6,7], suggest that the majority of troglobites and stygobites have not yet been described, with a prevalence of known macroscopic subterranean animals concerning the meiofauna, i.e., with a size between 0.06 and 1 mm [8].

The dissolution of carbonate rocks, such as limestone, sandstone, and dolomite, is caused by dissolved $CO_2$ in infiltrating water, which is diffused only in the shallower (i.e., up to 5 m below ground) fractured strata of the rock outcrops because the infiltrating runoff loses most of the $CO_2$ with depth. However, in rock outcrops with tectonic faults and sub-vertical fractures, the high downward-flow velocity of water can retain acidic properties for depths of more than 100 m, causing intense weathering and cavities [9].

The food web in karst groundwater habitats is based on organic compounds transported via runoff infiltration that are nutrients for hypogean animals, microorganisms, and fungi [10,11]. Hypogeous and groundwater biomes are characterized mainly by primary consumers, scavengers, and predator species owing to the absence of sunlight. Groundwater biocenosis often comprises rare endemic species because of the strong confined underground habitats [12]. Because troglobites and stygobites cannot survive in different habitats, the epikarst may consist of microhabitats that host specialized species [13]; these are highly vulnerable to changes in habitat conditions because of global warming. Climate change effects, rather than invasive species introduction, could pose major threats to stygofauna in epikarst freshwater. Some stygofauna species living in the fractured aquifers of Puglia, South Italy are endangered by habitat fragmentation and degradation owing to wastewater infiltrations [14]; they could rapidly decline because of changes in the seasonal groundwater flow and temperature regimes, increase in salinity due to a local sea-level rise, and more frequent and intense droughts and floods [15]. Moreover, in specific habitat conditions, such as in the case of the continuous input of organic matter, stygoxene could have favorable trophic conditions with metabolic rates higher than those of stygobiont taxa which may be more endangered [16,17].

A recent investigation, Blowes et al. [18], addressing the geography of biodiversity change in marine, terrestrial, and freshwater assemblages, shows that the average assemblage of global biodiversity and species richness is not changing because of both trends in the gain and loss of species at the local scale. Moreover, they suggest that biodiversity loss depends on the context of climate change and taxa location. Thus, species replacement and the rapid compositional change in biomes are ubiquitous and spatially structured, with local increases or decreases of up to 20% per year. The local compositional reorganization of biodiversity disagrees with its richness and is highest in the oceans.

Furthermore, according to Castellarini et al. [19], local changes in stygofaunal assemblages are correlated with habitat quality, such as elevation, rock hydraulic conductivity, and physical boundaries. Harpacticoids (subphylum: Crustacea, subclass: Copepoda) are found in various groundwater habitats worldwide, which have a typically limited dispersion capacity, and could serve as local indicators of habitat quality and variety [20].

The present study aims to determine the prospects of imminent species replacement and subsequent rapid compositional change in biodiversity caused by the impacts of climate change at a local scale in the karstified groundwater of cave and well habitats in Murgia and Salento, South Italy. Particularly, we proposed a new method to visualize 3D maps of the resilience of stygobitic species at local sites that are suitable for a quantitative comparison of different biodiversity hotspots. The quantitative estimation of the impacts of global warming on individual species of local biodiversity hotspots is important to test the efficacy of measures in managing plans to contrast with biodiversity loss, especially in habitats where species assemblages are highly sensitive to changes in groundwater velocity, salinity, temperature, dissolved oxygen, and $p$H.

## 2. Materials and Methods

We examined the species distribution of some assemblages monitored between 2018 and 2021 in four sites of karstic aquifers in Puglia and studied the morphological and physiological traits of the stygofauna species. Variations in traits were then associated with the degree of exposure and resilience (or vulnerability) of each sampled taxon to the change in habitat water temperature, $p$H, salinity, hydraulic conductivity, and pressure gradient to predict possible compositional adaptations and reorganizations of the local stygofauna

assemblage in response to the groundwater temperature and salinity increases caused by climate change [17]. Acclimation to thermal stress periods can be critical for the aquatic taxa in general and endemic groundwater species in particular [21], owing to the toxic accumulation of metabolic anaerobic end-products in animal tissues when they cannot be maintained below the control threshold. Peck et al. [22] reported the resilience of 14 marine species (limpet, brachiopod, bivalve mollusk, ascidian, starfish, urchin, brittle star, nemertean, anemone, gastropod, and amphipod) with increasing water temperature. The results suggest that the maximum upper-temperature tolerance limit of species, i.e., the physiological limit for survival, significantly decreased during acclimation periods of over one month. Furthermore, the resilience to selective processes that affected the trait distribution of the sampled species was investigated using the factor analysis [23].

### 2.1. Sampled Habitats

Samples of stygofauna were collected from four sampling sites in Puglia, Southern Italy (Figure 1). Monitoring wells where stygofauna were not retrieved are not shown in Figure 1. These stygofauna sites were considered representative of karstified groundwater in Murgia and Salento, which have different latitudes, elevations, and physicochemical conditions (Table 1).

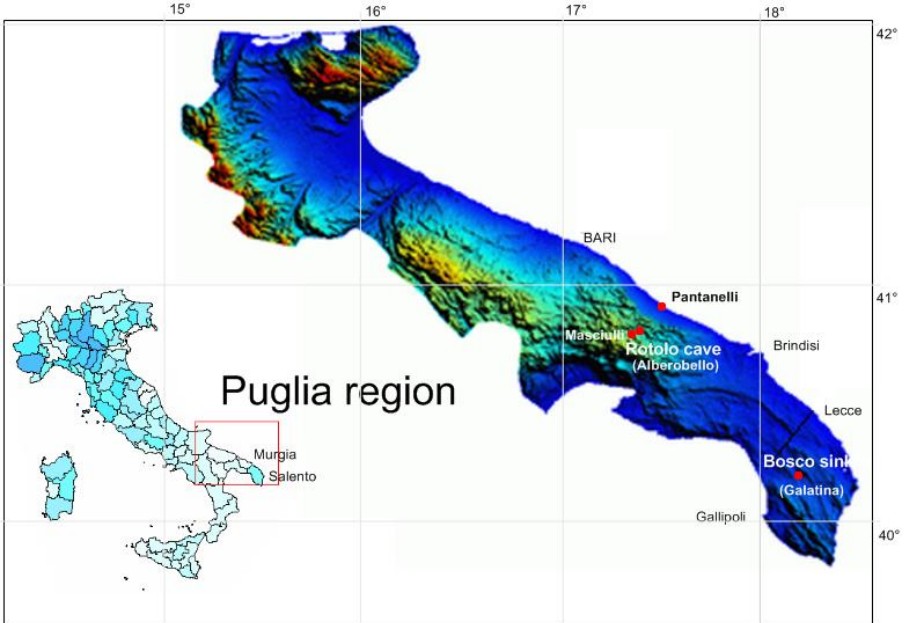

**Figure 1.** Stygofauna sampling locations, Rotolo cave and Bosco sinkhole, in Murgia and Salento, Southern Italy. The color of map (LANSAT) is associated with the soil elevation.

Rotolo cave was discovered recently and has characteristic maze-type cave developments from 305 m above sea level to over 40 m below sea level [24,25]. There are no geological heterogeneities between the initial conduit network of natural recharge (i.e., epikarst) and deep karst environments [26]. Owing to the advanced stage of the limestone rock karstification process, the karst groundwater table and perched aquifer in the epikarst are separate water bodies. Larger conduits and sub-vertical fractures allow rapid water infiltration and lateral inflow transmission from dolines into the conduit system and groundwater. Thus, surface runoff infiltration vanishes, and only a negligible amount of surface water can percolate via the fissures of the vadose zone in the groundwater. The sub-horizontal water flow in conduits of the perched aquifer (epikarst) was intercepted at different depths of the Rotolo sub-vertical shaft, forming temporary and permanent ponds or small lakes that are appropriate for stygofauna habitats.

**Table 1.** Sampling sites of regional stygofauna in Puglia, Southern Italy and monitored environmental conditions (average values).

| Sampling Locations | Longitude (N) | Latitude (E) | Hydraulic Conductivity (cm/s) | Distance from Sea (km) | Flow Gradient (%) | pH | Temperature (°C) | Electric Conductivity (mS/cm) | $O_2$ (mg/L) | Flow Head (m a.s.l.) | Soil Elevation(m a.s.l.) |
|---|---|---|---|---|---|---|---|---|---|---|---|
| Masciulli (Alberobello, BA) | 40° 49′ 20″ | 17° 15′ 04″ | 0.15 | 11.3 | 0.31 | 7.6 | 15.7 | 0.65 | 5 | 35 | 305 |
| Rotolo cave (Alberobello, BA) | 40° 49′ 30″ | 17° 15′ 08″ | 0.16 | 11 | 0.32 | 7.8 | 15.1 | 0.58 | 5.4 | 35 | 304 |
| Pantanelli (Monopoli, BA) | 42° 53′ 40″ | 17° 23′ 04″ | 0.2 | 0.2 | 0.05 | 8.1 | 11.0 | 8.3 | 7 | 0.1 | 0.5 |
| Bosco sinkhole (Noha, Galatina, LE) | 40° 10′ 1.5″ | 18° 10′ 15″ | 0.29 | 13.4 | 0.02 | 7.6 | 15.0 | 0.752 | 6 | 2.5 | 75 |

Sampling periods were 25 March–29 April 2018 at the Bosco sinkhole and 15 May–19 July 2021 at the other sites. The sampled habitats at Rotolo cave are temporary and permanent pools of epikarst, drip water from stalactite concretions, perched groundwater in the epikarst, and deep groundwater below the water table. The walls of water pools were scraped using the frame of nets, which caused a suspension of sediments and specimens in the water. The water containing sediments and specimens was immediately collected using manual nets with fine meshes (60 μm or 250 μm), and the water and substrate samples were transferred to 250 mL plastic bottles. Groundwater was investigated at depths of up to 20 m below the water table using a Cvetkov net (mesh size: 60 μm) [27], which has a metallic ring and sampling chamber at the bottom to retain the stygofauna and sediment samples. We used syringes in the sediments of water table shores to sample interstitial water with suspended matter and traps at the bottom of pools and boreholes. Furthermore, one borehole (Pantanelli) and six pumping wells, including the Masciulli well (see Figure 1) and five other wells close to the Rotolo cave were monitored in Murgia groundwater during the summer of 2021. From each pumping well, 100 L of groundwater was filtered through a 60 μm calibrated mesh net to collect the concentrated sediments into 250 mL bottles. The Cvetkov net was used in the borehole sampling. At Bosco, the sub-vertical karst conduit of a sinkhole conveys rainfall runoff into a small cave where the groundwater water table is 75 m below the ground. Groundwater was sampled using traps and a Cvetkov net. Sediments and stygofauna collected using nets or traps were gently washed with freshwater, fixed in 70% alcohol, and stored in a mobile refrigerator to ensure optimal temperature (5–10 °C) and darkness during sample transportation. Stygofauna were identified and classified in laboratory using a Leica MZ9.5 (Milan, Italy) stereoscope (www.spachoptics.com). The collected specimens were preserved in 75% ethyl alcohol solution and morphologically identified following the methods of Watson et al. [28]. Specifically, Harpacticoida and Cyclopidae were examined at a magnification between ×100 and ×400, and identified by Dussart [29,30] and Dussart and Defaye [31]. Amphipoda and Mysidacea were identified at a magnification between ×16 and ×40 following Messouli et al. [32], Nouvel et al. [33], Ruffo (1947) [34], Thorp and Rogers [35], and Wellborn et al. [36].

*2.2. Assessment of Increases in Groundwater Temperature and Salinity*

In a review of papers studying the increases in the salinity and temperature of groundwater owing to climate change, we may assess possible 2050 impacts on the investigated taxa at the Murgia and Salento sampling sites. Following the Intergovernmental Panel on Climate Change (IPCC) [37] projections of increasing global sea surface temperature under the actual negative scenario of higher global $CO_2$ emissions, we predicted an average value of less than 2 °C during 2050, i.e., an increasing rate of 0.05–0.06 °C per year. The

Rotolo groundwater temperature measured between 2017 and 2021 at 2 m depth using a multi-parametric probe (OTT ecology 800; www.ott.com) increased from 15.12 to 15.13 °C.

For the habitat salinity, recent IPCC 21st- and 22nd-century projections of mean global sea-level rise indicate an increase in saltwater intrusion in coastal aquifers, considering the increased melting rates of glaciers and ice caps globally, the effect of ocean thermal expansion, and changes in the quantity of stored land water. The resultant local Adriatic Sea level rise predictions match the global sea-level projections, with an increasing rate of 8.8 mm sea level/y. Sea intrusion into Murgia and Salento coastal aquifers was further assessed by Masciopinto and Liso [15] by interpolating local sea-level rise data since 2000. The estimated local sea-level rise has led to the prediction of seawater intrusion overshoots of 1.5 and 2.4 m in Murgia and Salento aquifers by 2050, respectively, with a corresponding increase in freshwater salinity of approximately 0.01 and 0.02 g/L.

## 3. Results

### 3.1. Resilience of Stygofauna to Climate Change

The details of bio-speleological samples of stygofauna collected from spring 2018 to summer 2021 from the Bosco sinkhole and Rotolo cave groundwater habitats are shown in Table 2. The identification of the sampled specimen "Cyclopidae C6" is incomplete, because of the inadequate data on the morphological traits of the adult females to clarify the dichotomy of specimens even at high magnification, suggesting the necessity of confirmation via further bio-speleological groundwater sampling in Rotolo cave.

Table 2 shows the results of 2018–2021 samplings in the four wells near the Rotolo site. The Hutchinsonian shortfall, i.e., the scientific gap in species tolerance to changes in abiotic factors, and the Raunkiæran shortfall, which concerns the unknown scientific aspects of the species traits, are suitable to describe scientists' difficulties in describing the factors affecting meiofauna in stabilizing species populations. Nevertheless, databases of functional species traits have helped us to predict the effects of climate change on the habitats of studied ecosystems and determine the potential of aquatic species for dispersal or the colonization of new groundwater habitats, similarly to Kano and Kase [38] and Gonzalez et al. (2017) [39] works dealing with gastropods and worms.

The ecological status shown in Table 2 was estimated via the species database interrogation of the classified species given by the Italian Committee of the International Union for Conservation of Nature (IUCN) (http://www.iucn.it/categorie.php) via the CKmap [40]. The IUCN Committee defined 11 categories of the ecological statuses of Italian species based on systematic criteria and extinction probabilities according to the available ecological data of the considered species. The CK-map checklist of the Italian faunal species [41], updated with achievements given by the specific project conducted by Ruffo and Stoch [42] of the Civic Museum of Natural History of Verona (Italy), provided the IUCN ecological estimations shown in Table 2 for any sampled species.

In the present study, we extended the IUCN criteria to determine the local resilience score (LRS) (Figure 2) of each sampled species to climate change in the studied habitats. For this, detailed information on species morphological and physiological traits, such as specimen size, endemism, dispersion capacity, trophic role, reproduction rate, salinity tolerance, *p*H, and tolerance to temperature changes, was investigated based on accurate inspection of collected stygofauna and literature. The LRS ranges from 1 to 10, where 1 is the most-endangered species, as explained in the following specific taxa descriptions. The IUCN conservation status had a prevalent weight in the LRS attributions.

**Table 2.** Results of the bio-speleological sampling of stygophiles (Sf), stygoxenes (Sg), and stygobites (Sb) in groundwater habitats of the Bosco sinkhole, Rotolo cave, and wells.

| Order | Species | 2018–2021 | Category | Ecological Status Concern | Adult/Young | Fem./Male Length (mm) | Male/Female | Endemic Degree | Coro Type | Trophic Role | Temp. Tolerance | Salinity Tolerance |
|---|---|---|---|---|---|---|---|---|---|---|---|---|
| Crustacea Copepoda Harpacticoida | H1 | Ponds of epikarst in Rotolo cave | Sf | Low | 22/0 | 0.6–0.68/0.58 | 11/11 | Low | EU | Gatherer | High | High |
| | H2 | Shaft outflow in Rotolo cave | Sb | | 36/0 | 0.42 | - | High | Italy | - | Low | Low (fresh water) |
| | H3 | Aquifer (Masciulli well) | Sg | Low | 1/0 | 0.7/0.5 | 0/1 | Low | Olartic | Gat. | High | High |
| Crustacea Copepoda Cyclopoida | C1 | Aquifer (Bosco) | Sb | Vulnerable | 7 | 0.62–0.64/0.50-0.9 | - | High | Med sea | Gat. | Low | Strong |
| | C2 | Ponds and aquifer in Rotolo cave | Sg | Low | 67/5 | 0.9–1.4/1 | 37/30 | Low | Olartic | Pred./gat | Very high | High |
| | C3 | Ponds and aquifer in Rotolo cave | Sf | Low | 4/0 | 0.7–1/0.7–0.9 | 3/1 | Low | EU | Gatherer | Quasi low | High |
| | C4 | Aquifer (Bosco) | Sg | Low | 1 | 0.9–1/0.7–0.9 | 0/1 | Low | Asia-EU | Gat. | High | High |
| | C5 | Aquifer (Pantanelli) | Sf | Low | 1/12 | 0.9–1.6/0.8–1.0 | 1/0 | Low | Olartic | Pred./gat. | Very high | High |
| | C6 | Ponds in epikarst of Rotolo cave | Sb | Low | 4 | - | 4/1 | - | - | - | High | High |
| Amphipoda | A1 | Aquifer (Bosco) | Sb | Endangered | 6/1 | 1.5–2.0 | - | High | Med Sea | Gat. | High | Quasi low |
| | A2 | Aquifer (Rotolo cave) | Sb | Quasi threatened | 1/0 | 2.8–4.7 | 1 | High | Italy | Gat. | - | Low (fresh water) |
| Mysidacea | M1 | Aquifer (Bosco) | Sb | Endangered | 9 | 6.5–13 | 9 | High | Italy | Fit./ Saproph age | High | High |

H1 Elaphoidella elaphoides elaphoides (Chappuis, 1924); H2 *Parastenocaris cf orcina* Chappuis, 1938; H3 *Bryocamptus pygmaeus* (Sars G.O., 1863); C1 *Metacyclops stammeri* Kiefer, 1938; C2 *Diacyclops bisetosus* (Rehberg, 1880); C3 *Paracyclops imminutus* Kiefer, 1929; C4 *Paracyclops fimbriatus* (Fischer, 1853); C5 *Diacyclops bicuspidatus bicuspidatus* (Claus, 1857); C6 Cyclopidae gen 1 sp 1 (under study); A1 *Salentinella gracillima* Ruffo, 1952; A2 *Hadzia minuta* Ruffo, 1947; M1 *Spelaeomysis bottazzii* Caroli, 1924.

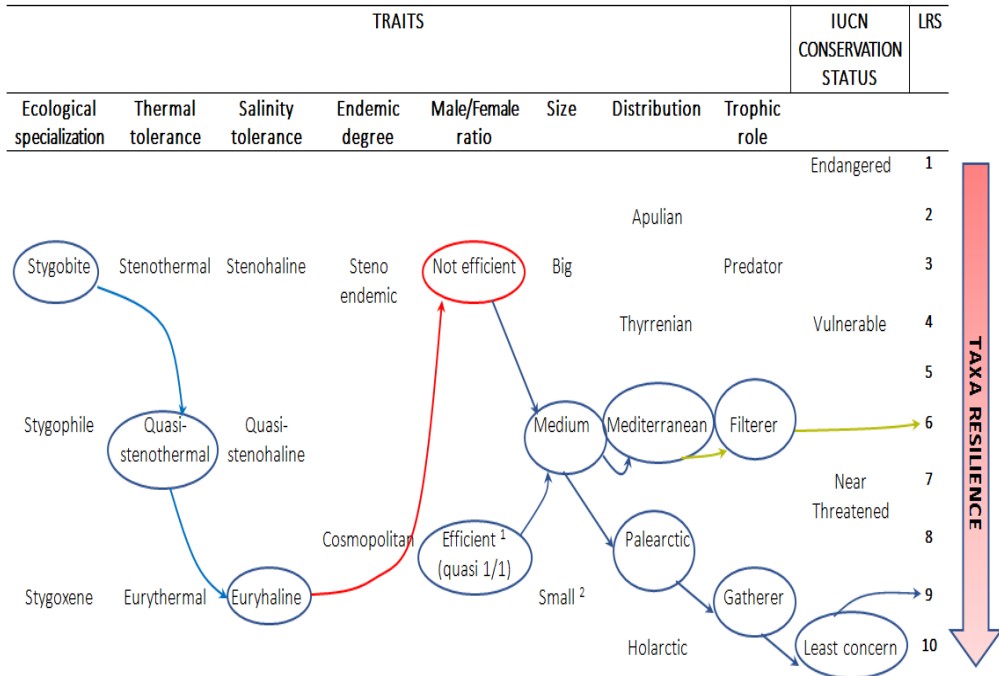

1 - An M/F ratio of about 1/1 indicates a high potential for reproductive efficiency and, potential taxa ability to recolonize habitats after environmental disturbances;

2 - A smaller size of animals allows efficient species dispersion throughout the fracture apertures (< 1 mm) and cavities of karst habitats.

**Figure 2.** Decisional flow visualization of local resilience score (LRS) attributions to the sampled stygofauna at Murgia and Salento groundwater.

H1: *Elaphoidella elaphoides elaphoides* (Chappuis, 1924)

It is a stygophile that is widely distributed in both surface and groundwater. It is a euryhaline, eurythermal, and non-endemic species with an IUCN status of least concern. The collected specimens were adults with an M:F ratio of 1:1, indicating an overall efficient fertility rate. The small size of specimens (0.6–0.68 mm for females and 0.58 mm for males) compared with the cavity size of karstic conduits (>1 cm) and fissures (>1 mm, on average) together with euriecian and generalist traits of this species justify its wide holoendemic and Palaearctic distribution. These harpacticoids are gatherers and not limited by resources, such as bio-colloids and dissolved organic compounds in groundwater. Therefore, we assigned an LRS of 9.

H2: *Parastenocaris* cf. *orcina* Chappuis, 1938

It is a hypogean and stygobiont species with a low tolerance to water temperature and salinity changes, and it is a stenoendemic species with a narrow distribution area (Tyrrhenian). However, its small size (0.42 mm) may be advantageous for its dispersal ability. Therefore, we assigned an LRS of 5.

H3: *Bryocamptus pygmaeus* (Sars G.O., 1863)

It is an epigean and a stygoxene. It is a scarcely specialized, eurythermal, euryhaline, and non-endemic species with an IUCN status of least concern. It is a cosmopolitan gatherer with large spatial distributions (Holarctic) and is frequently observed in several aquatic habitats, including wastewater [43]. The small size (0.70 mm for females; 0.50 mm for males) facilitate its dispersion. Therefore, we assigned an LRS of 8.

C1: *Metacyclops stammeri* Kiefer, 1938

It is a gatherer hypogean and stygobiont species. It is euryhaline and endemic to Mediterranean with an IUCN status of vulnerable. Although it was not collected during sampling in Rotolo cave, owing to the small size (0.5–0.9 mm) of specimens and its vulnerable status, we assigned an LRS of 4.

C2: *Diacyclops bisetosus* (Rehberg, 1880)

It is an epigean and a stygoxene species. It is cosmopolitan, scarcely specialized, eurythermal, euryhaline, and non-endemic species with an IUCN status of least concern. We collected 67 adult and juvenile specimens from the Rotolo cave, with a size of 0.9–1.4 mm for females and 1.0 mm for males, and an M:F ratio of 1.2:1, indicating an overall efficient fertility rate. Thus, we assigned an LRS of 10.

C3: *Paracyclops imminutus* Kiefer, 1929

It is an epigean and a stygophile species. It is a moderately specialized, euryhaline, quasi-stenothermal, and non-endemic (European) gatherer species with an IUCN status of least concern. The specimens were small (<1 mm), which facilitates species dispersion in karstic aquatic habitat environments. However, owing to its moderate resilience to water temperature changes, we assigned an LRS of 8.

C4: *Paracyclops fimbriatus* (Fischer, 1853)

It is an epigean and a stygoxene and a generalist species, frequently observed in most habitat types of running epigean waters and karstic aquatic/benthic habitats of Asia and Europe. It is highly eurythermal and euryhaline and has an IUCN status of least concern. The gatherer specimens were <1 mm in size. Thus, we assigned an LRS score of 9.

C5: *Diacyclops bicuspidatus bicuspidatus* (Claus, 1857)

It is an epigean and a stygophile species. It is highly eurythermal and euryhaline and has an IUCN status of least concern. It is a ubiquitous holoendemic species with a wide distribution (Holarctic), with specimens smaller than 1.6 mm. Thus, we assigned an LRS score of 10.

C6: Cyclopidae gen 1 sp 1

It is a new hypogean copepod species collected during two samplings on May 5 and 17 July 2021 from the epikarst conduit outflow, approximately 40 m above the karst groundwater piezometric surface in Rotolo cave. The taxonomic identification of the specimens is in progress; thus, the provisory name Cyclopidae gen 1 (i.e., unknown genus 1) sp 1 (i.e., unknown species 1) has been given for this new stygobitic fauna. We assigned an LRS score of 6, which is higher than that of the stygobitic cyclopoid C1 sampled from the sinkhole. This is because the new copepod could have an expansive range in groundwater habitats with environmental quality better than that of a sinkhole.

A1: *Salentinella gracillima* Ruffo, 1952

It is a foraging hypogean and stygobiont species endemic to the Murgia and Salento karstic groundwater habitat. It prefers oligohaline water, and the size (1.5–2 mm) may limit its dispersion. It is classified as an endangered species by the IUCN; thus, we assigned it a low LRS of 2.

A2: *Hadzia minuta* Ruffo, 1947

It is a hypogean and stygobiont species, which prefers stenohaline water and is endemic to Italy. The species is confined to highly fragmented groundwater habitats owing to its large size (2.8–4.7 mm). Because it is classified as "near threatened" by the IUCN, we assigned the lowest LRS of 1 for this amphipod species.

M1: *Spelaeomysis bottazzii* Caroli, 1924

It is a hypogean and stygobiont species. It is endemic to Apennine Mountains in Apulian karstic groundwater habitats. Moreover, it is eurythermal and euryhaline and is classified as "endangered" by the IUCN. This species tolerates light and is found in partially contaminated habitats [14]. The size of this species is 6.50–13 mm, and it was collected in fragmented coastal aquifers with large spatial distribution. Thus, we assigned an LRS of 3.

### 3.2. Mathematical Model of Stygofauna Resilience

The link between biodiversity loss because of habitat deterioration and functional traits of sampled stygofauna was investigated using statistical factor analysis. The morphological, functional, and behavioral features of stygobite specimens in niches or fractures of groundwater habitats can be distinct from those of typical epigean species adapted to

greater spatial volumes. Thus, we excluded *Paracyclops fimbriatus* (C4), *Diacyclops bisetosus* (C2), and *Bryocamptus pygmaeus* (H3) in the sampled assemblage from the factor analysis. All monitored abiotic conditions and local species resilience scores were used for factor analysis. Correlation analysis was performed using SPSS for testing relationships between several habitat variables and the assigned resilience scores. The factor analysis result, i.e., the matrix of the correlation coefficients, was rotated to highlight the maximum correlation links among the considered variables. Specifically, we observed a moderate positive correlation of the LRS with the groundwater temperature (T) (54%), salinity (0.56), and dissolved oxygen (DO) (0.56) and a negative correlation (−0.44) of that with the groundwater velocity via hydraulic conductivity, K (m/s), of the rock and the piezometric gradient, H (m). Groundwater *p*H was strongly correlated with the electrical conductance (EC) (and distance from the seacoast, ds) and temperature (EC–T) factor (0.81) and moderately with DO (0.44). Figure 3 shows the spatial three-dimensional visualization of the three factors, DO, EC–T, and K, which may explain the mechanistic relationship between climatic changes in groundwater temperature and salinity and the adaptive modification of stygofauna assemblage in the investigated groundwater habitats.

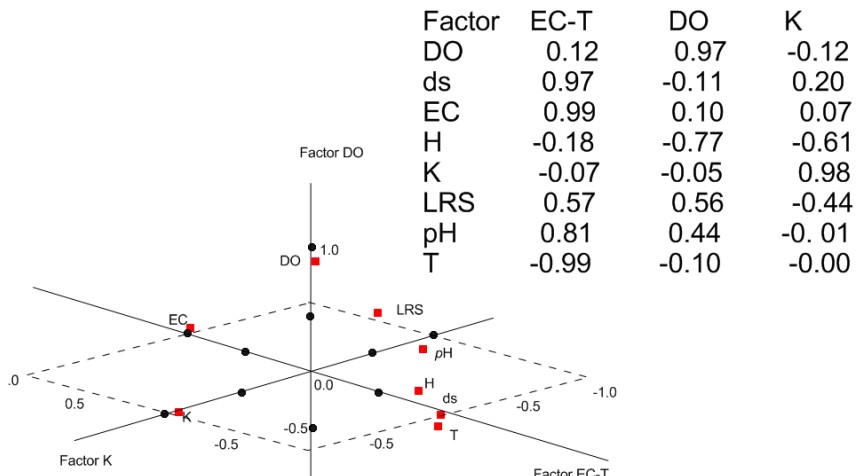

| Factor | EC-T | DO | K |
|---|---|---|---|
| DO | 0.12 | 0.97 | -0.12 |
| ds | 0.97 | -0.11 | 0.20 |
| EC | 0.99 | 0.10 | 0.07 |
| H | -0.18 | -0.77 | -0.61 |
| K | -0.07 | -0.05 | 0.98 |
| LRS | 0.57 | 0.56 | -0.44 |
| pH | 0.81 | 0.44 | -0. 01 |
| T | -0.99 | -0.10 | -0.00 |

**Figure 3.** Three-dimensional visualization of abiotic habitat factors of electrical conductance (EC) (and distance from the seacoast, ds) and temperature (T) (EC–T), dissolved oxygen (DO), hydraulic conductivity (K) (and pressure head, H), and rotated correlation matrix showing correlations between LRS and EC–T, DO, and K factors.

A successive factor analysis was then applied to exclude LRS values to rearrange abiotic habitat conditions in independent variables or factors that can be used for a regression model suitable for simulating local stygofauna resilience. The results provide two factors or independent variables, the first factor mainly correlated with groundwater flow and salinity, and the second factor correlated with DO and *p*H. Furthermore, SPSS results save the standardized predicted values of these independent factors (or variables) for each independent species. The nonlinear regression between the assigned LRS, as defined by the functional analysis of traits, and corresponding factor values (Table 3) defined the proposed resilience model.

Table 3 shows that the LRS values assigned to the Murgia and Salento groundwater assemblages may have different values owing to the change in *X* or *Y* estimations using the monitored parameters in the specific sampled habitat, as shown in the following equations:

$$X = 0.28\text{ ds} + 0.26\text{ T} - 0.25\text{ EC} - 0.21\,p\text{H} + 0.131\text{ K} - 0.05\text{ DO} - 0.02\text{ H} \tag{1}$$

$$Y = 0.13\text{ ds} + 0.05\text{ T} - 0.02\text{ EC} + 0.09\,p\text{H} + 0.40\text{ K} + 0.31\text{ DO} - 0.54\text{ H} \tag{2}$$

The regression coefficients of Equations (1) and (2) are obtained from the SPSS output. The X and Y values (Table 3) correspond with the variable values (Table 1), that is, groundwater flow (K and H), salinity, temperature, DO, and *p*H.

To define the equation of the resilience model of groundwater stygofauna of the Murgia, and groundwater in the context of climate change, we performed LRS (, i.e., Z) predictions using a surface best-fit of the assigned values of Z as a function of X and Y, as shown in Table 3. We applied TableCurve 3D® (v.4.0.01; Systat Software, https://systatsoftware.com/; accessed on 29 June 2022) for the surface best-fit. The results provided the following surface model of the species resilience (SSR) equation

$$LRS = 6.5 - 5.0\,X - 1.42\,Y^2 \tag{3}$$

Which yielded a standard error of 2.5 for Z values in Table 3. A map of the stygofauna surface resilience model is depicted in Figure 4.

**Table 3.** Species conservation scores (Z) owing to different functional adaptive traits in the same groundwater habitats and factors (*X*) and abiotic environmental conditions (*Y*) monitored between 2018 and 2021.

|  | LRS (Z) | X (EC-T) | Y (H-K) |
|---|---|---|---|
| H1 | 9 | −0.011 | 0.629 |
| H2 | 5 | 0.061 | −1.232 |
| C1 | 4 | 0.584 | 0.816 |
| C3 | 7 | 0.330 | −1.346 |
| C5 | 10 | −2.565 | 0.212 |
| C6 | 6 | −0.012 | 0.613 |
| A1 | 2 | 0.776 | 0.770 |
| A2 | 1 | 0.330 | −1.346 |
| M1 | 3 | 0.507 | 0.884 |

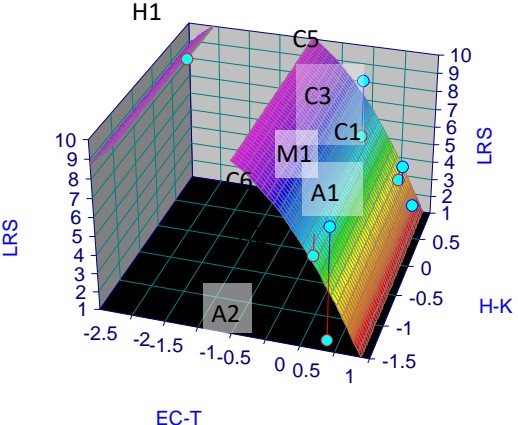

**Figure 4.** Stygofauna surface resilience (SSR) map: $r^2 = 0.63$, fitted standard LRS error = 2.1, significance level $F_{stat} = 5.1$.

However, because stygophiles have an intermediate ecology compared to that of stygobites and epigean fauna, the above statistical procedure was repeated to focus only on stygobitic species. Thus, in the first step, SPSS factor analysis yielded only two factors with a rotated matrix that highlighted correlation coefficients showing a strong correlation (94%) of LRS with DO and a minor correlation (27%) with salinity and groundwater flow velocity factors compared to those with the water temperature factor (16%). The salinity factor links the EC, K, ds, H, and T in the monitored habitat. Groundwater *p*H was correlated with DO, whereas water temperature was more correlated with EC–K than with DO. Figure 5 shows a two-dimensional visualization of the EC–K and DO factors.

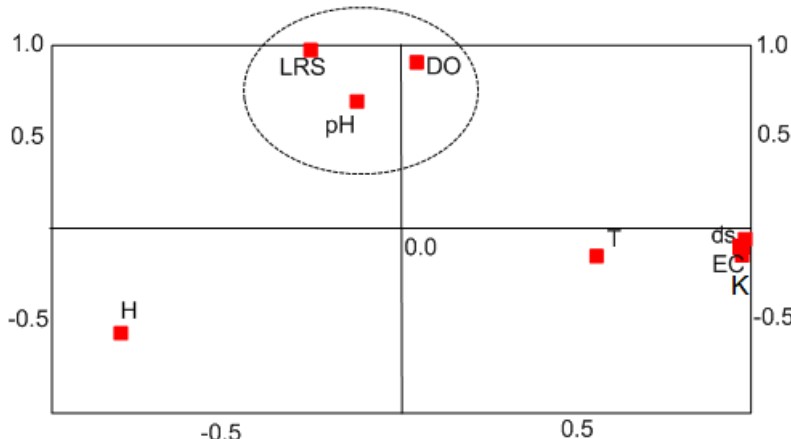

**Figure 5.** Visualization of factors EC–K and DO, and coefficients showing a high correlation between LRS, DO, and $p$H.

Subsequently, an additional factor analysis was performed by excluding the known assigned LRS values. The SPSS output provided the independent components $X_s$ and $Y_s$ of the stygobiont species for the resilience model. TableCurve 3D model of surface fitting of the assigned LRS using the corresponding $X_s$ and $Y_s$ values for each species provided the resilience equation:

$$\ln(LRS) = -2.9 + 5.7\, X_s - 2.7\, X_s^3 + 2.6\, e^{-X_s} + 0.5\, Y_s^3 \tag{4}$$

Figure 6 depicts the map of stygobitic resilience given by Equation (4).

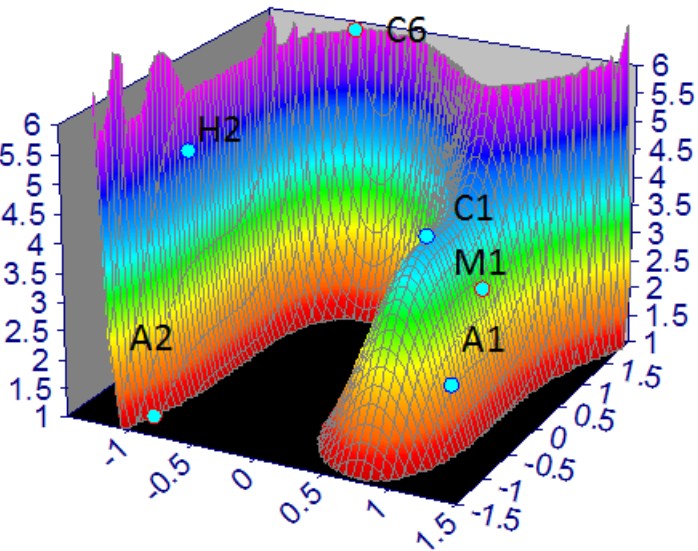

**Figure 6.** Map of stygobiont resilience ($r^2 \cong 1$; standard error = 0.01; $F_{stat} = 2.6 \times 10^4$) sampled from Murgia and Salento groundwater (2018–2021).

### 3.3. Projections of Stygofaunal Resilience for 2050

The stygofauna resilience model (4) can quantitatively predict any effect caused by habitat alterations in groundwater stygofauna species at a local scale, owing to changes in water temperature, salinity, and related variables, such as dissolved oxygen, $p$H, and flow velocity. Changes in environmental conditions can be caused by global warming, anthropogenic activities, or combined effects. Predictions can be performed by updating $X_s$ and $Y_s$ abiotic components using the resilience model in Equation (4).

In the present study, the prediction of climate change impact for 2050 on the Murgia and Salento stygobitic assemblage was made by incorporating updated components in $X_s$

and $Y_s$, such as the increase in the 2021 groundwater temperature and a specific conductance of up to 2 °C and 0.04 mS/cm, respectively, and neglecting other related changes of *pH*, DO, H, K, and ds for simplification. Higher values of T and EC were assigned to the deep-sampled groundwater habitats. The prediction of the SSR is plotted in Figure 7 via the 3D map of the surface fitting scores predicted from the same surface fitting Equation (4).

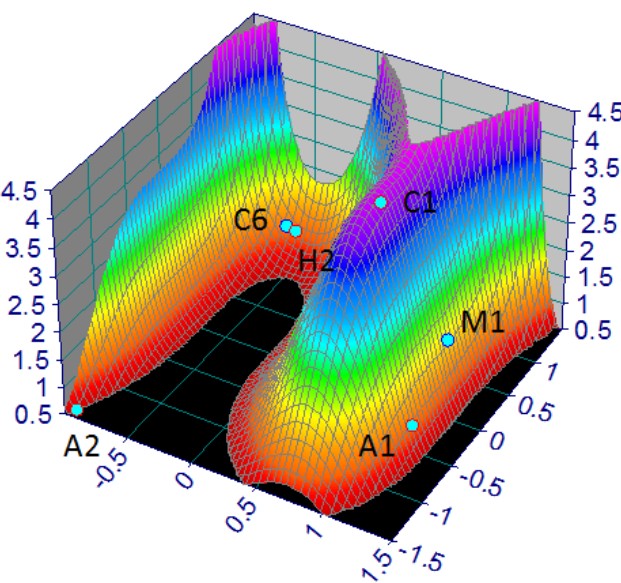

**Figure 7.** Prediction of SSR drawdown of Murgia and Salento groundwater for 2050 owing to the increases in groundwater temperature of up to 2 °C and conductance of up to 0.04 mS/cm.

Model Equation (4) was confirmed using TableCurve 3D as the best equation for fitting the LRS forecasts. The 2050 resilience reductions for each stygobitic species shown in Figure 8 predicted significant effects on the resilience of *Parastenocaris* cf. *orcina* (Chappuis 1938) (H2) (80%) and newly retrieved copepod classified as Cyclopidae gen 1 sp 1 (C6) of epikarst groundwater, other than three stygobitic taxa (about 50%) of deep groundwater. *Metacyclops stammeri* (Kiefer 1938) (C1) showed a negligible impact of resilience to global warming.

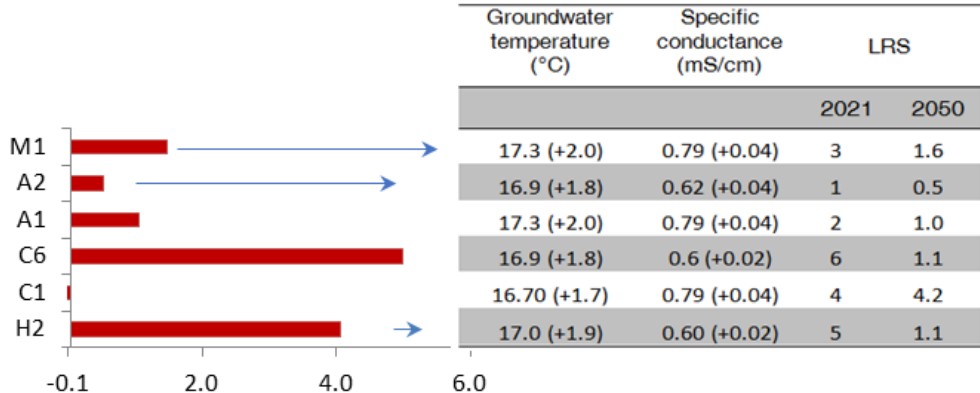

| | Groundwater temperature (°C) | Specific conductance (mS/cm) | LRS | |
|---|---|---|---|---|
| | | | 2021 | 2050 |
| M1 | 17.3 (+2.0) | 0.79 (+0.04) | 3 | 1.6 |
| A2 | 16.9 (+1.8) | 0.62 (+0.04) | 1 | 0.5 |
| A1 | 17.3 (+2.0) | 0.79 (+0.04) | 2 | 1.0 |
| C6 | 16.9 (+1.8) | 0.6 (+0.02) | 6 | 1.1 |
| C1 | 16.70 (+1.7) | 0.79 (+0.04) | 4 | 4.2 |
| H2 | 17.0 (+1.9) | 0.60 (+0.02) | 5 | 1.1 |

**Figure 8.** Prediction of 2021 reductions for 2050 in stygofauna taxa resilience in Murgia and Salento groundwaters because of climate change, leading to groundwater temperature and specific conductance increases of up to 2 °C and 0.04 mS/cm.

## 4. Discussion and Conclusions

The proposed method provides quantitative 3D maps of the resilience of stygobitic species at local sites. Today, 3D maps of species resilience surfaces are rare in the specialist

literature, and the new modeling approach of the present study can provide significant insights into the context of global climate change.

These resilience surface maps are suitable for quantitative visualization and comparison of different biodiversity hotspots. SSR provided different effects of anthropogenic activities and/or climate change for the investigated species assemblages. The quantitative estimation of the impacts of global warming on individual species of local biodiversity hotspots can be suitable for testing the efficacy of measures in managing plans to contrast biodiversity loss, especially in habitats where species assemblages are highly sensitive to changes in groundwater velocity, salinity, temperature, dissolved oxygen, and *p*H.

Model results confirm that epikarst-specialized species of investigated habitats are more endangered than deep groundwater fauna, as suggested by Pipan et al. [13], because the former are highly vulnerable to changes in biotic and abiotic habitat conditions as a consequence of global warming. A multivariate statistical analysis performed by Shapouri et al. [44] indicated that stygofauna assemblages sampled in a coastal aquifer of Portugal significantly varied with location, suggesting that changes in the composition of stygofauna assemblages are indicators of seawater intrusion into coastal aquifers.

Improvements can be made to the proposed model by including LRS estimations of metabolic effects, i.e., mortality, on the investigated taxa assemblages in groundwater using relationships derived from the results presented by Castaño-Sánchez et al. [45]. These authors quantified the resilience of groundwater-adapted crustaceans endemic to Australia for increases in both water temperature and salinity using laboratory experiments. The results show a better resilience of syncarids and cyclopoids compared to harpacticoids, which have a 10% mortality rate in groundwater temperature or salinity over 21.5 °C or 0.6 g/L, respectively.

The limitations of the SSR model, owing to the uncertainty posed by the applied statistical methods and approximations owing to the best-fit procedure, can be reduced by increasing the sample size. This also means that a variety of sampling sites can increase the significance of model outputs and SSR maps. The weak significance of SSR best-fit observed is notable when stygophile and stygobite species are considered in the same factor analysis.

The proposed SSR model positively depicted the species resilience of the Murgia and Salento groundwater stygofauna assemblages collected during bio-speleological sampling performed in groundwater at a depth of 264 m, epikarst at the Rotolo cave, and Bosco sinkhole groundwater at a depth of 72 m. Twelve species, six stygobites, three stygophiles, and three stygoxenes were classified. A possibly new stygobitic copepod species was identified in Rotolo cave, as the morphology of the retrieved specimens differed from that of the extant Italian genera of this family. This specimen was observed in water samples collected during two different samplings of the same epikarst conduit outflow from Rotolo cave. This observation of new species is not unusual in the biodiverse Murgia and Salento groundwater karst habitats. A new stygobiont species *Stygocyclopia badinoi* sp. nov. (Family: Pseudocyclopiidae) was recently discovered in the Zinzulusa cave (Lecce, Italy) in Salento [46], which hosts a total of 25 stygobiont species [47]. These reports of new species could be, as suggested by Blowes et al. [18], taxa replacements in the caves of the Murgia and Salento groundwater habitats owing to climate change impacts. This increase in biodiversity may only partially balance the deficit caused by potential losses in biodiversity assemblages of Murgia and Salento groundwater taxa because many previously reported stygobitic species, such as *Metaingolfiella mirabilis* (Ruffo 1969), *Niphargus* gr. *orcinus* (Joseph 1869), *Nitokra* af. *psammophila* (Noodt, 1952), and *Nitocrella stammeri* (Chappuis, 1938), have not been recorded since 1991.

**Author Contributions:** Conceptualization and methodology, C.M. and A.T.D.C.; software, C.M.; validation, formal analysis, C.M.; investigation, A.T.D.C.; resources, C.M.; data curation, C.M. and A.T.D.C.; writing—original draft preparation, writing—review and editing, C.M. and A.T.D.C.; visualization, C.M.; supervision, A.T.D.C.; project administration, C.M.; funding acquisition, C.M. All authors have read and agreed to the published version of the manuscript.

**Funding:** The research was funded by the project Grotta Rotolo 2 of Puglia Region (I) (Regione Puglia: D.G.R. n. 806/2019), Department of Mobility, Urban Quality, Public Works, Ecology and Landscape (Dr. A. RICCIO—Regione Puglia: D.G.R. n. 806/2019).

**Data Availability Statement:** Manuscript data are reported in Tables 1 and 2.

**Acknowledgments:** We thank M. Parise, coordinator of the project activity, University of Bari, D.M.P. Galassi and her team (B. Fiasca, M. Di Cicco and I. Vaccarelli) of the University of L'Aquila, Italy), who identified a new species of copepod in the groundwater habitats of the epikarst; the speleologist groups of GASP! and Avanguardie for carrying out sampling in the Rotolo cave and Bosco sinkhole, respectively; and I.S. Liso, student of Bari University for the sampling of wells.

**Conflicts of Interest:** The authors have no conflict of interest to declare.

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
