# Peer review of "Modeling Stygofauna Resilience to the Impact of the Climate Change in the Karstic Groundwaters of South Italy"

_water, doi:10.3390/w14172715_

Round 1

Reviewer 1 Report

I have two minor comments and one major comment.  The minor comments are 

line 128 Please add a reference for the Cvetkov net

Line 144 When I first read this paragraph I thought it belonged in the discussion.  However, I think it is in the right place but its context, i.e., a justification of how to assess climate change, isn't immediately clear. Please add a sentence or change the subheading.

The major comment

line 221.  The actual criteria for giving a particular score to a species is unclear.  The IUCN website lists the categories but there is a missing link with the actual assignment of numbers.  Given their importance in the rest of the analysis, this needs fixing.

Author Response

The authors appreciated the time and impressive work made by reviewer #1 to improve this manuscript. Thank you

Authors’ replays (AR) to reviewer comments or suggestions are reported below in italics.

Reviewer 1:

I have two minor comments and one major comment. 

The minor comments are:

1) line 128 Please add a reference for the Cvetkov net

AR: Citation was provided: Cvetkov L., 1968. Un filet phréatobiologique. Bulletin de l’Institut de Zoologie et Musée, Sofia, 22: 215-219.)

2) Line 144 When I first read this paragraph, I thought it belonged in the discussion.  However, I think it is in the right place but its context, i.e., a justification of how to assess climate change, isn't immediately clear. Please add a sentence or change the subheading.

AR: The authors have revised the subheading title to “Assessment of increase of groundwater temperature and salinity”. Moreover, we include the following explanation,

“In a review of papers studying the increase of salinity and temperature of groundwater owing to climate change, we may assess possible 2050 impacts on the investigated taxa at the Murgia and Salento sampling sites.”

The major comment

line 221.  The actual criteria for giving a particular score to a species is unclear.  The IUCN website lists the categories but there is a missing link with the actual assignment of numbers.  Given their importance in the rest of the analysis, this needs fixing.

AR: A new picture has been included to explain the decisional flow for attributions of the taxa local resilience score (LRS).

Reviewer 2 Report

The paper ”Modeling stygofauna resilience to the impact of the climate change in the karstic groundwaters of South Italy” represents an interesting article. I have several comments:

Line 39: air-filled niches [citation needed].

Lines 86-95: This paragraph belongs to Material & Methods. Instead, please comment on the novelty and importance of the study.

Lines 103-105, Figure 1: is there a code of colors? Why use different shades of blue on the small map of Italy, and why use blue/green/red on the other map?

Line 119: the values in Table 1 are averages? Please clarify. Also: pH without italics; mg/L instead of mg/l

Lines 97-143: Sampled habitats:

-Please describe the sampling methods used for Masciulli and Pantanelli. The authors focused on Rotolo cave and Bosco sinkhole

- ”Furthermore, one borehole and six pumping wells at different distances from the Adriatic Sea coast were  monitored in Murgia during the summer of 2021”: please describe these sampling sites too, and show them on the map from Figure 1

- what was the sampling period?

- ” The collected specimens were preserved in 75% ethyl alcohol  solution and morphologically identified following the methods of Watson et al. ”: it is unclear at this point: only harpacticoid copepods were identified? What about the other groups included in Table 2? Cyclopidae, Amphipoda, Mysidaceae?

Lines 144-179: Assessment of climate change impacts:

-much information presented here must be included in ”Discussions” (at least the lines from 150 to 165). Please relocate and rearrange the text, and focus in this part on materials and methods used.

-the first use of IPCC must include the non-abbreviated name

Line 180: The whole chapter ”2.3. Resilience of stygofauna to climate change” must be relocated in ”Results”. Only the paragraph between lines 216 and 221 is suitable for ”Material and Methods”

Line 188: Table 2:

-Masciulli and Pantanelli are not included. Or are these two locations labelled as ”the wells”? Please clarify.

-I cannot see in the table the exact location of identification of species (Rotolo? Bosco? well #1?). Is this a less important detail?

Line 206: the two papers cited here (30 and 31) deal with gastropods and worms. How can these databases be reliable for the present study?

Lines 310-332: What is ”EC-T”? What is ”ds”? Please explain better in the text.

Lines 333-335, Figure 2: Please explain the abbreviations used in the figure and in the figure caption. The values presented in Figure 2 are redundant with the text above

Lines 451-453: Figure 7: please delete the maps, since they were shown in Figures 5 and 6

Lines 454-491: Discussions: Please add the paragraphs from ”Material and methods”. Also, please compare your results with similar ones (if any) and stress the importance (and novelty?) of the present study in the context of global climate change.

Line 510: 2019 a

Line 521: 2019 b

Author Response

The authors appreciated the time and impressive work made by reviewer #2 to improve this manuscript. Thank you.

The authors' replies are in the attached file

Round 2

Reviewer 1 Report

The authors have addressed my concerns.